# Atroposelective Formal [2 + 5] Macrocyclization Synthesis for a Novel All-Hydrocarbon Cyclo[7] *Meta*-Benzene Macrocycle

**DOI:** 10.3390/molecules29143363

**Published:** 2024-07-17

**Authors:** Chao Gao, Hongchen Li, Jing Zhao, Lulu Bu, Mei Sun, Jingrui Wang, Gang Tao, Longde Wang, Li Li, Guilin Wen, Yunhu Hu

**Affiliations:** 1School of Chemistry and Materials Engineering, Huainan Normal University, Huainan 232038, China; jingzhao@hnnu.edu.cn (J.Z.); lulubuvip@163.com (L.B.); hssm202106@163.com (M.S.); jingruiwang2003@163.com (J.W.); gangtao0103@163.com (G.T.); ldwang72@126.com (L.W.); lli@mail.ustc.edu.cn (L.L.); guilinwen@126.com (G.W.); 2CNOOC Institute of Chemicals & Advanced Materials, Beijing 102209, China; lihc19@tsinghua.org.cn

**Keywords:** Suzuki–Miyaura coupling, atropisomerism, racemization, adaptive chirality

## Abstract

A novel axially chiral all-hydrocarbon cyclo[7] (1,3-(4,6-dimethyl)benzene (**CDMB-7**) was designed and synthesized using atroposelective[2 + 5] cyclization through Suzuki–Miyaura coupling. **CDMB-7** adopts an irregular bowl-like shape with *C_2_* symmetry and exhibits two diastereoisomers in its crystallographic structure. The conformational isomers of **CDMB-7** racemates remain stable at high temperatures (393 K). High-performance liquid chromatography (HPLC) confirmed that a single chiral isomer will spontaneously undergo racemization within 30 min at room temperature. This finding opens up possibilities for achieving adaptive chirality in all-hydrocarbon cyclo[7] m-benzene macrocycles.

## 1. Introduction

Atropisomerism is widely observed in pharmaceutical molecules [1,2,3], chiral natural products [4,5,6], chiral ligands [7,8,9], and specific macrocyclic molecules [10]. The phenomenon primarily arises from restricted rotation around a molecular axis (single bond) [11,12], facilitating the chiral separation of individual enantiomers [13]. Such separation presents opportunities for creating novel chiral drugs and functional materials. However, delving into the intricacies of isomerization within supramolecular chemistry remains a formidable challenge. The macrocycles highlighted in various studies, including crown ethers [14,15], carbon nanobelts [16,17], nanorings [18,19,20], cuppyrroles [21], triptycene derivatives [22], calixarenes [23,24,25], cyclobenzenes [26,27,28,29,30], superhelicenes [31,32,33], etc., exhibit considerable intrinsic or potential isomerization values, which have a profound impact on their synthesis and functional applications. Given this, it is essential to harness the diverse performance of these macrocycles in chiral self-recognition, detection, characterization, and fluorescence [34,35,36].

Cyclo-*meta*-phenylenes (**CMPs**), characterized by a representative macrocycle [37,38], have attracted widespread interest due to their simple composition and unique structure. So far, due to the lack of involved synthesis and modification strategies, few all-hydrocarbon as-prepared CMPs and related derivatives **CDMB-8** [39] (Figure 1A) have been reported. Herein, we build a new, all-hydrocarbon cyclo[7] (1,3-(4,6-dimethyl)benzene) (**CDMB-7**) that exists in *C_2_* symmetry. Two non-enantiomeric isomers of CDMB-7 can be generated simultaneously, and still maintain the stability of the configuration at high-temperatures (393 K), without causing conversion due to external stimuli.

## 2. Results

The synthesis of **CDMB-7** is shown in Figure 1B. Briefly, the Suzuki–Miyaura coupling “[2 + 5]” cyclization between boronylated dimer **1** [40] and dibrominated pentamer **2** afforded the target macrocycle **CDMB-7** in a yield of 26%. Dibrominated pentamer **2** was generated from the reported work [39]; then, with the varieties of chiral ligands involved and through controlling the process of asymmetric cyclization [41,42,43], unfortunately, the obtained pure products are still racemic mixtures. Its structure was subsequently confirmed by single crystal X-ray diffraction.

The [Pd], base, and solvent effects in the cyclization from **1**, **2**, and **3** are explored for the reaction optimization (Table 1). The highest 26% yield is achieved using Cs_2_CO_3_ under Ar and PhMe (Note: the reaction solvents are all analytical reagents (ARs)).

The structures of **3** are fully characterized via (^1^H, ^13^C, COSY, NOESY, ROESY) NMR, high-resolution mass spectra (HMRS, MALDI-FTICR), and further confirmed by single crystal X-ray diffraction analysis. In particular, only **3** showed racemates containing two diastereoisomers, which implied that the [2 + 5] cyclization is a highly *diastereoselective* reaction.

It is applied to preliminary single chiral isomer exploration by adding additional chiral ligands under the optimized condition. This showed that using (*R*)-(+)-2,2′-Bis(diphenylphosphino)-1,1′-binaphthyl or (*S*)*-(+)*-2,2′-Bis(diphenylphosphino)-1,1′-binaphthyl, the yields are 43% and 40%, respectively (Table 2, entry 1, 2). However, the addition of other chiral phosphine ligands did not have a significant promotional effect on the yield, as shown in Table 2. Hence, it suggests that the (*S*)-binaphthyl ligands positively influence the enhancement of the reaction yield by generating a bidentate coordination reaction intermediate in the same orientation as **2**. Thus, both (*R*)-BINAP and (*S*)-BINAP can produce the racemic form of **CDMB-7** with a high yield.

Moreover, the HPLC-confirmed target macrocycles, obtained after the reaction of all chiral ligands involved, were found to be racemic. It indicates that the introduction of chiral ligands did not effectively regulate the chirality of products. The rigidity of fragments **1**, **2** was probably the main factor in generating the single chiral isomer.

## 3. Discussion

The ^1^H NMR spectrum of **CDMB-7** (produced through the above synthetic process) recorded at 273 K in tetrachloroethane-*d*_2_ (TCE-*d*_2_) is characterized by six signals for the benzene ring protons that appear in a 1:4:3:2:2:2 ratio (Figure 2a). Seven groups of signals are seen (appearing as three overlapping peaks) corresponding to the meso-methyl groups. In addition, 21 signals are also seen in the ^13^C NMR spectrum (Appendix A, **ESI**). The findings mentioned above help us prove that **CDMB-7** adopts a fixed structure with *C_2_* symmetry in TCE-*d_2_*.

More evidence for the macrocycle formation with *C_2_*-symmetry came from crystal diffraction analysis [44,45,46,47,48]. The crystal was grown via slow evaporation in DCM/CH_3_CN (1:2, *v*/*v*). The crystal structure showed that ***C_2_*-CDMB-7** has a foldable curved irregular “bowl-like” conformation with the seven meso-dimethyl benzene units. And the diameter of **CDMB-7** in different views is 6.762 Å to 9.250 Å (Appendix A, **ESI**). Interestingly, two non-enantiomeric isomers coexist as racemates in the crystal (Figure 3b).

According to the analysis of NMR, the two-dimensional spectrum revealed that the benzene ring interactions between adjacent hydrogen–hydrogen and remote hydrogen–hydrogen in the macrocyclic molecule **CDMB-7** are relatively weak (Figure 2b). The HSQC and HMBC two-dimensional spectrum have not been detected. Thus, it confirmed that there is almost no interaction between the inside carbon and hydrogen atoms of **CDMB-7** macrocycle, also explaining the macrocycle structure’s existence as individual units. The variable temperature ^1^H NMR showed some shifts in specific signals seen upon heating at 283–393 K temperature range, the whole spectroscopic peak of the NMR recorded at 283 K was retained (Figure 2c). It indicates that the conformer of ***C_2_*-CDMB-7** exists in a de-symmetrization shape-persistent nature at high temperatures.

## 4. Materials and Methods

We have achieved the separation of the single chiral isomer with *diastereoselectvity* of **CDMB-7**, generating peaks with an area ratio of 1:1 in chromatographic analysis column **AD-H** (mainly filled with silica surface covalently bonded cellulose-tri (3,5-dimethylphenylcarbamoyl) and the factor of chiral separation reaches 2 in HPLC (Figure 3a). Then, it successfully separated two single conformational isomers of the diastereoselective ***C_2_*-CDMB-7** (separate yield < 5%, *ee*: 85%, 83%) using n-hexane and methanol (99/1, *v*/*v*) as the mobile phase through **AD** chromatographic preparation column at 30 °C.

However, it was further investigated when the two diastereoselective separated chiral isomer solutions were allowed to stand at room temperature for 30 min; the respective samples were injected again, and the signals of the two sets of chromatographic peaks showed consistency with the determination results of the racemic mixture in HPLC.

The initial crystallographic data obtained through chiral mode X-ray diffraction analysis (Figure 3b) showed two diastereoselective isomers of **CDMB-7** rapidly reaching racemization in solution. It is postulated that the absence of strong polar or bulky substituents, other than the methyl group, in the molecular structure of **CDMB-7** contributes to this conversion. This construction results in a shallow energy barrier for the configurational interconversion of its chiral isomers. Solvent interaction further facilitates this process, allowing the molecules to attain an energetic equilibrium easily.

Consequently, the single chiral species transform into their enantioselective images (mirror isomer), culminating in forming a stable racemic mixture in **CDMB-7** solid phase. The findings of this research suggest that the nature of the functional group substitutions in **CDMB-7**, as well the selection of solvent, are likely the primary determinants influencing its enantioselective reactivity. On the other hand, the meso-dimethyl multi-aromatic ring units of ***C_2_*-CDMB-7** and its irregular twisted conformation have the characteristic of available enantiomerization for racemization.

This study establishes a foundation for further investigation of the self-adaptive chirality of ***C_2_*-CDMB-7**. ***C_2_*-CDMB-7** can responsively alter its chiral conformation in reaction to diverse external stimuli, such as pressure, solvents, and guest molecules. This adaptive shift in chiral configuration or conformation elicits distinctive chiral responses. Furthermore, we try to synthesize a novel chiral pharmaceutical derived from the **CDMB-7** scaffold, tailored to interact with the humors, blood, or additional biological mediators within the organism. This interaction aims to effectuate a conformational shift in the target drug molecule, enhancing its lesion-specific therapeutic potency, thereby, eradicating associated pathologies. Moreover, we strive to attain the precise modulation of **CDMB-7** analog adaptive chiral entities to guarantee their conformational transition is exclusive to particular solvent systems. Such specificity would enable these molecules to discriminate selectively against distinct ions, chiral diminutive molecules, or chiral moieties in a solution, thus augmenting the enantioselectivity of chiral recognition.

## 5. Conclusions

In conclusion, we have demonstrated a novel axially chiral cyclo[7] (1,3-(4,6-dimethyl)benzene (**CDMB-7**), synthesized through atroposelective Suzuki–Miyaura [2 + 5] cyclization in 43% yield. This protocol offers an economical and straightforward approach for directly synthesizing all-hydrocarbon odd-numbered multi-aromatic macrocycles. Analytical studies, including X-ray, NMR, and HPLC, identified that the atroposelectivity primarily originates from the rigidity of the as-prepared fragments, and the conformation of ***C_2_*-CDMB-7** can undergo spontaneous “racemization”. These insights pave the way for further investigations into the self-adaptive chirality of ***C_2_*-CDMB-7** and its potential application in dynamic chiral recognition, detection, and luminescence.

## Figures and Tables

**Figure 1 molecules-29-03363-f001:**
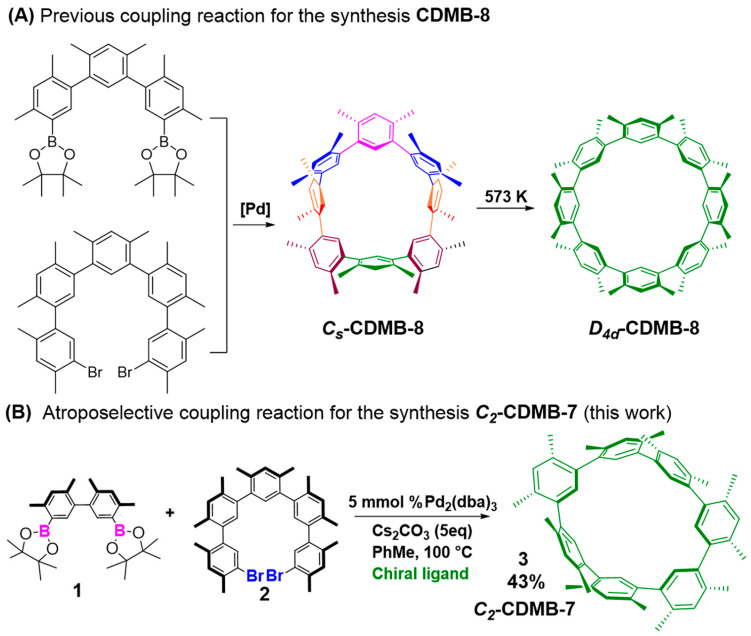
(**A**) Previous examples of **CDMB-8** [39] synthesis; (**B**) Suzuki–Miyaura coupling [2 + 5] cyclization for ***C_2_***-**CDMB-7** synthesis (this work).

**Figure 2 molecules-29-03363-f002:**
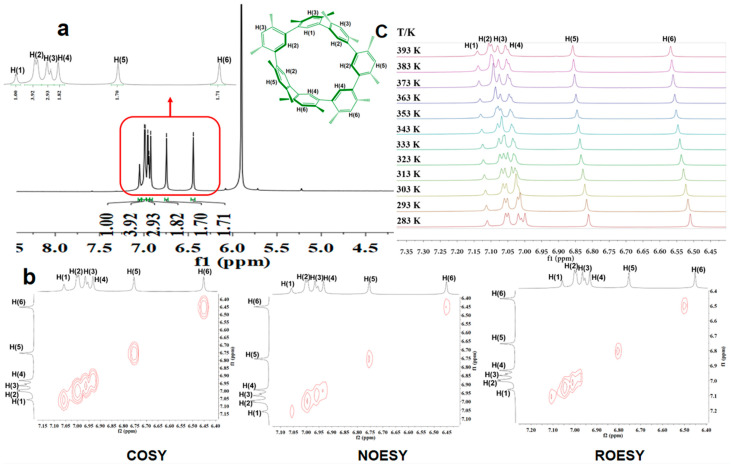
(**a**) ^1^H NMR (273 K, 500 MHz) of benzene ring protons in **CDMB-7** (5.00 mM); (**b**) expansion of the temperature dependent ^1^H MR (500 MHz) spectra of ***C_2_*-CDMB-7** (5.00 mM) in TCE-*d_2_*; (**c**) COSY, NOESY, or ROESY two-dimensional spectra of ***C_2_*-CDMB-7** (5.00 mM).

**Figure 3 molecules-29-03363-f003:**
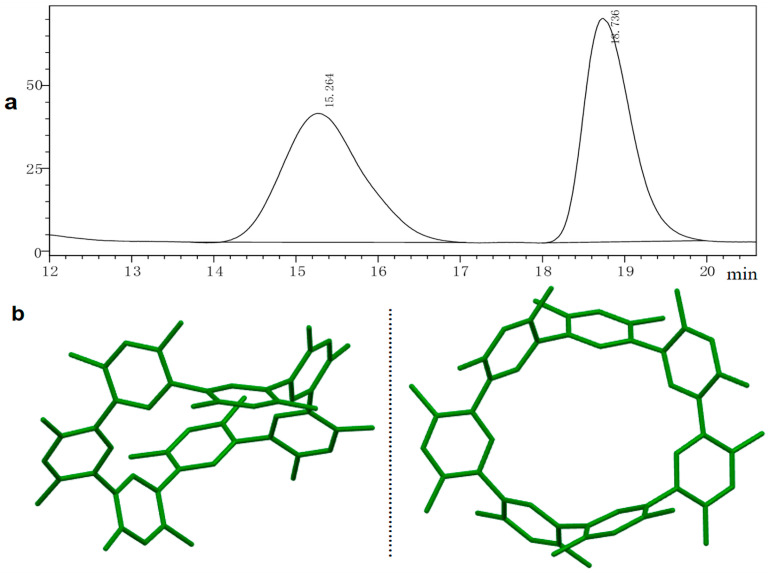
(**a**) HPLC peaks of ***C_2_*-CDMB-7** in chromatographic column **AD-H** (2.0 × 10^−4^ M, n-hexane/isopropanol, 95/5, *v*/*v*, 25 °C); (**b**) ***C_2_*-CDMB-7** original crystal racemates under the chiral mode of X-ray crystal diffraction.

**Table 1 molecules-29-03363-t001:**
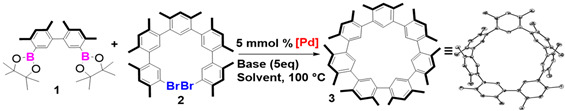
Optimization of reaction conditions ^a^.

Entry	[Pd]	Base	Solvent	Yield(%) ^b^
1	Pd_2_(dba)_3_	Cs_2_CO_3_	PhMe	26
2	Pd(dppf)_2_Cl_2_	Cs_2_CO_3_	PhMe	25
3	Pd[(PPh_3_)]_4_	Cs_2_CO_3_	PhMe	ND ^c^
4	Pd(OAc)_2_	Cs_2_CO_3_	PhMe	<5
5	PdCl_2_	Cs_2_CO_3_	PhMe	ND
6	Pd_2_(dba)_3_	Cs_2_CO_3_	DMF	ND
7	Pd_2_(dba)_3_	Cs_2_CO_3_	DMSO	<5
8	Pd_2_(dba)_3_	Cs_2_CO_3_	CH_3_CN	Mess ^d^
9	Pd_2_(dba)_3_	Cs_2_CO_3_	THF	18
10	Pd_2_(dba)_3_	Cs_2_CO_3_	DMA	ND
11	Pd_2_(dba)_3_	K_2_CO_3_	PhMe	15
12	Pd_2_(dba)_3_	CH_3_COOK	PhMe	22
13	Pd_2_(dba)_3_	NaOH	PhMe	Mess
14	Pd_2_(dba)_3_	CsF	PhMe	Mess
15	Pd_2_(dba)_3_	*^t^*BuONa	PhMe	ND

^a^ Reaction conditions: **1**, **2** (0.15 mmol, 0.075 M, 1 equiv.), [Pd] (5 mmol %, 3.75 × 10^−3^ M), base (0.75 mmol, 0.375 M, 5 equiv.), solvent (2.0 mL), 100 °C, Ar, 12 h. ^b^ Isolated yields. ^c^ Not detected. ^d^ Mess: an inseparable mixture.

**Table 2 molecules-29-03363-t002:** Exploration of chiral phosphine ligand ^a^.

Entry	Ligand ^a^	Yield(%) ^b^
1	(*R*)-(+)-2,2′-Bis(diphenylphosphino)-1,1′-binaphthyl	40
2	(*S*)-(+)-2,2′-Bis(diphenylphosphino)-1,1′-binaphthyl	43
3	(*R*)-(+)-TolBINAP	18
4	(*R*)-(+)-SEGPHOS	15
5	(*R*)-(*R*)-JOSIPHOS	12
6	(*R*)-Ph-Garphos	<5
7	(*S*)-(*R*)-JOSIPHOS	23
8	(*S*)-DM-SEGPHOS	30
9	(*S*)-2,2′-Bis(di-3,5-xylylphosphino)-1,1′-binaphthyl	35
10	(*R*)-(+)-MOP	22

^a^ Reaction conditions: **1**, **2** (0.15 mmol, 0.075 M, 1 equiv.), [Pd] (5 mmol %, 3.75 × 10^−3^ M), base (0.75 mmol, 0.375 M, 5 equiv.), ligand (15 mmol %, 1.12 × 10^−2^ M), solvent (2.00 mL), 100 °C, Ar, 12 h. ^b^ Isolated yields.

## Data Availability

CCDC 2330139 accessed on 1 February 2024 contain the supplementary crystallographic data for this paper. These data can be obtained free of charge via www.ccdc.cam.ac.uk/data_request/cif, or by emailing data_request@ccdc.cam.ac.uk, or by contacting The Cambridge Crystallographic Data Centre, 12 Union Road, Cambridge CB2 1EZ, UK; fax: +44-1223-336033.

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
