# Peer review of "Atroposelective Formal [2 + 5] Macrocyclization Synthesis for a Novel All-Hydrocarbon Cyclo[7] *Meta*-Benzene Macrocycle"

_molecules, 2024, doi:10.3390/molecules29143363_

Round 1

Reviewer 1 Report

Comments and Suggestions for Authors

Comments

The manuscript describes the synthesis of the previously unknown all-hydrocarbon macrocycle, consisting of seven m-dimethylbenzene moieties. Synthesized compound (CDMB-7) belongs to a scarce type of hydrocarbons which are perspective as receptors of non-polar organic molecules. The structural features of CDMB-7 have been detected by NMR spectroscopy and X-Ray analysis. The manuscript is suitable for the publication in Molecules, but the following corrections are to be done (significant revision is required).

1.     The error in the title of manuscript and SI should be corrected: “Cycly[7] m-Benzene” should be changed for “Cyclo[7] m-Benzene”.

2.     Lines 37-45: When describing the known cyclo-meta-phenylenes, the authors refer to the works published in 1960s (ref. 37,38), but no more recent references are provided (see for example, dx.doi.org/10.1021/jo501903n, https://doi.org/10.1039/C5SC03807C  etc). The reference 39 (Chem. Commun. 2019, 55, 3701–3704) must appear for compound CDMB-8 which is given in Fig.1, a, and in the text (line 40). It is strange that the authors mention this work (Chem. Commun. 2019, 55, 3701–3704, ref. 39) only as a reference to the starting compounds 1 and 2 for the synthesis of CDMB-7. Moreover, compd 1 in ref. 39 is absent. Also, the error in the journal name in ref 39 should be corrected (“Chem. Comcom”, lines 262-263)

3.     Phrase “gram scale synthesis of 3” which was obtained in 146 mg yield – not applicable (line 60)

4.     Unfortunately, the description of the composition of the obtained substance raises many questions. As it is pointed in the manuscript, CDMB-7 presents a mixture of two diastereoisomers as racemates (“Two non-enantiomeric isomers of CDMB-7 can be generated simultaneously” – are they diastereomers? lines 42-43). In this case, two sets of signals for two diastereomers should be observed in NMR spectra, but only one set is observed. Also, it is unclear, whether two pure diastereomers or two pure enantiomers were obtained by chromatographic separation? What process is observed after separation: racemization or interconversion of diastereomers? (section Discussion).

The following phases are not clear:

“In particular, 3 can showed racemates containing two diastereoisomers and another enantiomer pair was absent”, lines 63-64

“two non-enantiomeric isomers coexist as racemates in the crystal”, lines 99-100

It would be useful to present an image of the resulting stereoisomers of CDMB-7 and to elucidate relevant parts of the text.

5.     The phrase “It implied that the [2+5] cyclization has very high diastereoselectivity, even diastereospecificity” is incorrect. According to IUPAC gold book “A reaction is termed stereospecific if starting materials differing only in their configuration are converted into stereoisomeric products…” (https://goldbook.iupac.org )

6.     The term “axochiral” is not a general meaning, it should be changed to “axially chiral”

7.     DFT calculation for the observed inversion of conformation could be added to understand the thermodynamic parameters of this process.

8.     In view that both (R)-BINAP and (S)-BINAP can be used to obtain CDMB-7 with comparable yields, it makes sense to use this ligand in a racemic form

9.     Does the designation "mess" in the table 2 mean "an inseparable mixture"? It should be clarified

10.  Figures 1-3, and especially figure 2, as well as the figures in SI, are of poor resolution. It is difficult to realize spectral information from Figure 2.

11.  In the caption to fig. 2 “500 Hz” should be changed to “500 MHz”

12.  The experimental procedure for compound CDMB-7 should be included to Supporting Information or to the section Materials and Methods. Moreover, NMR spectra for  compound CDMB-7 should be checked. For example, the compound with 14 Me groups cannot contain three signals with one among them (δ 2.21 ppm) assigned to 7 protons. It seems that the integral intensity was not accurately measured.

Comments on the Quality of English Language

Minor editing of English language required, for example, "luminescen"

Author Response

  • Suggestion 1: The error in the title of manuscript and SI should be corrected: “Cycly[7] m-Benzene” should be changed for “Cyclo[7] m-Benzene”.

    Answer: Thanks for the careful check and kind suggestion. The “Cycly[7] m-Benzene” has changed into “Cyclo[7] m-Benzene” in the title of manuscript and SI.

    Suggestion 2: Lines 37-45: When describing the known cyclo-meta-phenylenes, the authors refer to the works published in 1960s (ref. 37,38), but no more recent references are provided (see for example, dx.doi.org/10.1021/jo501903n, https://doi.org/10.1039/C5SC03807C etc). The reference 39 (Chem. Commun. 2019, 55, 3701–3704) must appear for compound CDMB-8 which is given in Fig.1, a, and in the text (line 40). It is strange that the authors mention this work (Chem. Commun. 2019, 55, 3701–3704, ref. 39) only as a reference to the starting compounds 1 and 2 for the synthesis of CDMB-7. Moreover, compd 1 in ref. 39 is absent. Also, the error in the journal name in ref 39 should be corrected (“Chem. Comcom”, lines 262-263)

    Answer: Thanks for the kind suggestion. After careful consideration, we decided to change references [37,38] to the recent references (J. Org. Chem. 2014, 79, 9735–9739; Chem. Sci. 2016, 7, 896–904). The reference [39] is added in Fig.1, a, and in the text (line 40) of manuscript. Ref. 39 is only used to guide the synthesis of compound 2, and the description of compound 1 in line 52 has been deleted in latest manuscript. The journal name of ref. 39 has been changed to “Chem. Commun.”.

    Suggestion 3: Phrase “gram scale synthesis of 3” which was obtained in 146 mg yield – not applicable (line 60).

    .Answer: Thanks for the kind suggestion. The description of “gram scale synthesis of 3” which was obtained in 146 mg yield” in line 60 has been deleted in section.

    Suggestion 4: Unfortunately, the description of the composition of the obtained substance raises many questions. As it is pointed in the manuscript, CDMB-7 presents a mixture of two diastereoisomers as racemates (“Two non-enantiomeric isomers of CDMB-7 can be generated simultaneously” – are they diastereomers? lines 42-43). In this case, two sets of signals for two diastereomers should be observed in NMR spectra, but only one set is observed. Also, it is unclear, whether two pure diastereomers or two pure enantiomers were obtained by chromatographic separation? What process is observed after separation: racemization or interconversion of diastereomers? (section Discussion).

    The following phases are not clear:

    “In particular, 3 can showed racemates containing two diastereoisomers and another enantiomer pair was absent”, lines 63-64

    “two non-enantiomeric isomers coexist as racemates in the crystal”, lines 99-100

    It would be useful to present an image of the resulting stereoisomers of CDMB-7 and to elucidate relevant parts of the text.

    Answer: Appreciate so much for the careful check and kind suggestion. We can intuitively observe that in the original crystal (measured in chiral mode) of CDMB-7 (Fig. 1), a pair of diastereomers exist simultaneously (lines 42-43). X-ray  crystal diffraction confirmed that the product obtained is a mixture of racemic form (lines 99-100). Thus, the “mixture” showed only the signal response of racemate on 1H NMR spectra, while the two single chiral isomers were not observed. “In particular, 3 can showed racemates containing two diastereoisomers and another enantiomer pair was absent” means the two diastereomers does not exist in the isomers that present a mirror image of their respective structures (lines 63-64). And we will upload the original crystal for check.

    On the other hand, we get two pure diastereomers from chromatographic separation. The diastereomers undergo racemization, and part of their structures are transformed into enantiomers with mirror symmetry after separation.

    Fig. 1 original crystal structure measured in chiral mode of CDMB-7 in one view

    Suggestion 5: The phrase “It implied that the [2+5] cyclization has very high diastereoselectivity, even diastereospecificity” is incorrect. According to IUPAC gold book “A reaction is termed stereospecific if starting materials differing only in their configuration are converted into stereoisomeric products…” (https://goldbook.iupac.org ).

    Answer: Thanks for the careful check and kind suggestion. After careful consideration, we try to change the description of phrase “It implied that the [2+5] cyclization has very high diastereoselectivity, even diastereospecificity” into “It implied that the [2+5] cyclization is a stereospecific reaction”.

    Suggestion 6: The term “axially chiral” is not a general meaning, it should be changed to “axially chiral”.

    Answer: Thanks for the careful check and kind suggestion. We have changed to “axially chiral” in lines 13, 161.

    Suggestion 7: DFT calculation for the observed inversion of conformation could be added to understand the thermodynamic parameters of this process.

    Answer: Thanks for the kind suggestion. Theoretically, through the previous work ((J. Org. Chem. 2014, 79, 9735–9739), it is feasible to use the thermodynamic parameters calculated by DFT calculation to observe inversion of conformation process from single chiral isomer to racemates of CDMB-7. During the operation, we tried to cultivate single chiral isomer crystal of CDMB-7 in various ways, but unfortunately failed, and also failure to obtain the cif. Thus, it leads to the calculation of thermodynamic parameters for complete conformational inversion of monochiral isomer and racemate of CDMB-7 has not been successfully realized.

    Suggestion 8: In view that both (R)-BINAP and (S)-BINAP can be used to obtain CDMB-7 with comparable yields, it makes sense to use this ligand in a racemic form.

    Answer: Thanks for the careful check and kind suggestion. We assume that chiral ligands are mainly used to induce the formation of CDMB-7 single chiral isomer. Then the specific experimental results show that ligands (particularly (R)-BINAP and (S)-BINAP) positively influence the enhancement of the reaction yield of CDMB-7 racemic form and the yield is 38% under BINAP (no chiral) condition, a little effect for chiral induction. The description is in lines 70-74 in detail.

    Suggestion 9: Does the designation "mess" in the table 2 mean "an inseparable mixture"? It should be clarified.

    Answer: Thanks for the careful check. The "mess" in the table 1 has been clarified in line 80, and it means "an inseparable mixture".

    Suggestion 10: Figures 1-3, and especially figure 2, as well as the figures in SI, are of poor resolution. It is difficult to realize spectral information from Figure 2.

    Answer: Thanks for the careful check and kind suggestion. The resolution of Figures 1-3, as well as the figures in SI, has changed to 300 ppi. And we will upload the high resolution pictures for check.

    Suggestion 11: 11.  In the caption to fig. 2 “500 Hz” should be changed to “500 MHz”.

    Answer: Thanks for the careful check. The fig. 2 “500 Hz” has been changed to “500 MHz” in line 114.

    Suggestion 12: The experimental procedure for compound CDMB-7 should be included to Supporting Information or to the section Materials and Methods. Moreover, NMR spectra for compound CDMB-7 should be checked. For example, the compound with 14 Me groups cannot contain three signals with one among them (δ 2.21 ppm) assigned to 7 protons. It seems that the integral intensity was not accurately measured.

    Answer: Appreciate so much for the careful check and kind suggestion. The experimental procedure for compound CDMB-7 is added in page 4 of Supporting Information. About the compound CDMB-7 with 14 Me groups contain three signals with one among them (δ 2.21 ppm) assigned to 7 protons, it bases on the literature: “Aggregation of Complexes Coordinated with N,N′-Bis(5-alkylsalicylidene)ethylenediamine:1H NMR Peak Shifts and Paramagnetic Broadening Investigations (Bull. Chem. Soc. Jpn. 1989, 62, 45-50)” and “Enhancing the resolution of 1H and 13C solid-state NMR spectra by reduction of anisotropic bulk magnetic susceptibility broadening (Phys. Chem. Chem. Phys. 2017, 19, 28153--28162)” explain that the polymerization and rapid movement of molecular will cause NMR peak broadening excluding the problem of experimental parameters (shimming, etc.). The NMR peak broadening will lead to the concentration of multiple protons on Me groups on one signal peak, then no obvious signal splitting can be produced. CDMB-7 1H NMR were collected under 5.00 mM at 273 K. The experimental results show that the density of 5.00 mM was beneficial to promote the intermolecular polymerization of CDMB-7 monomer molecules, which led to the broadening of 1H peaks. And 273K is also to reduce the probability of rapid movement of CDMB-7 molecules in solution.

    Suggestion 13: Minor editing of English language required,for example, “luminescen”.

    Answer: Appreciate so much for the careful check and kind suggestion. The “luminescen” has changed into “luminescence” in line 171. And we have also check and corrected other sections in the manuscript.

  • Please see the attachment in detail.

Reviewer 2 Report

Comments and Suggestions for Authors

The authors report the preparation and characterization of a molecular ‘belt’ composed of seven aromatic rings. The particular compound is a regular 3,5-dimethyl substitution pattern, which appears to be a unique compound and a good example of high selectivity in synthesis in this area. The authors provide ample data for characterization, including spectroscopy and even X-ray crystallographic characterization. The data presented supports the formulation made. The presentation is clear, and the references made are highly appropriate. I have no concerns about the this study and its overall quality. I would recommend for publication with only very minor revision: The experimental conditions of data collection for the X-ray analysis should be updated including features like temperature, absorption correction, etc. These can be found in a cif check.

Comments on the Quality of English Language

English quality is good.

Author Response

  • Suggestion: The authors report the preparation and characterization of a molecular ‘belt’ composed of seven aromatic rings. The particular compound is a regular 3,5-dimethyl substitution pattern, which appears to be a unique compound and a good example of high selectivity in synthesis in this area. The authors provide ample data for characterization, including spectroscopy and even X-ray crystallographic characterization. The data presented supports the formulation made. The presentation is clear, and the references made are highly appropriate. I have no concerns about the this study and its overall quality. I would recommend for publication with only very minor revision: The experimental conditions of data collection for the X-ray analysis should be updated including features like temperature, absorption correction, etc. These can be found in a cif check.

    Answer: Thanks for reviewer careful check and kind suggestion. We have tried our best to seriously revised the manuscript on the basis of all the reviewer’s suggestions, including features like temperature, absorption correction is added in page 6 of SI.

  • Please see the attachment in detail.

Round 2

Reviewer 1 Report

Comments and Suggestions for Authors

In general, the authors took into account the comments and corrected necessary points, but some questions remain about the stereochemistry and NMR data of the compound CDMB-7

1. The authors’ answer on Comment 4 is not satisfactory. The situation with diastereomers and enantiomers was not elucidated and still rises many questions. Terms “diastereomer” and “enantiomer” are mixed and the sense rests vague.

2. Also (comment 5), the term “diastereospecificity” or “stereospecificity” is applied only “if starting materials differing only in their configuration are converted into stereoisomeric products”. In this work achiral starting compounds turn into mixtures of two diastereomers of product. This result corresponds to “diastereoselectvity” or “stereoselectvity

3. To the answer to comment 12: Signals broadening does not result in the changes of integral intensity. Thus you need simply to integrate all methyl groups not trying to integrate selected signals.

Author Response

  • In general, the authors took into account the comments and corrected necessary points, but some questions remain about the stereochemistry and NMR data of the compound CDMB-7

    Suggestion 1: The authors’ answer on Comment 4 is not satisfactory. The situation with diastereomers and enantiomers was not elucidated and still rises many questions. Terms “diastereomer” and “enantiomer” are mixed and the sense rests vague.

    Answer: Thanks for the careful check and kind suggestion. Compound CDMB-7 synthesized by template reaction was considered a mixture of two racemates. Although 1H NMR didn’t show two pairs of peaks by some reason unascertained (probably the structures of the two racemates are very same), but both HPLC and X-ray single crystal diffraction experiments proved 3 is a “racemates”. After careful consideration, we also modified the description of mixed “diastereomer” and “enantiomer” has been corrected in lines 62-64, 118, 122-123, 126, 130-132, 137-138.

    Suggestion 2: Also (comment 5), the term “diastereospecificity” or “stereospecificity” is applied only “if starting materials differing only in their configuration are converted into stereoisomeric products”. In this work achiral starting compounds turn into mixtures of two diastereomers of product. This result corresponds to “diastereoselectvity” or “stereoselectvity”.

    Answer: Thanks for the careful check and kind suggestion. Through the reported works (Angew. Chem. Int. Ed. 2019, 58, 13904; Molecules 2022, 27, 2847), we have corrected the description of phrase to “In particular, 3 only showed racemates containing two diastereoisomers, which implied that the [2+5] cyclization is a high diastereoselectvity reaction” in lines 62-64.

    Suggestion 3: To the answer to comment 12: Signals broadening does not result in the changes of integral intensity. Thus you need simply to integrate all methyl groups not trying to integrate selected signals.

    Answer: Appreciate so much for the careful check and kind suggestion. We have re-integrated all methyl groups on CDMB-7, and the 1H NMR description and spectra of CDMB-7 has been modified in page 5, 10 of SI. And we will upload the high resolution picture of 1H NMR for check.
